

**Investigating the response of China's surface ozone**
**concentration to the future changes of multiple factors**
Jinya Yang [1], Yutong Wang[1], Lei Zhang[1, 2], Yu Zhao[1, 2*]
1. State Key Laboratory of Pollution Control and Resource Reuse, School of
Environment, Nanjing University, 163 Xianlin Rd., Nanjing, Jiangsu 210023, China
2. Jiangsu Collaborative Innovation Center of Atmospheric Environment and
Equipment Technology (CICAEET), Nanjing University of Information Science and
Technology, Jiangsu 210044, China
*Corresponding author: Yu Zhao
Phone: 86-25-89680650; email: *yuzhao@nju.edu.cn*



## Abstract

Climate change and associated human response are supposed to greatly alter
surface ozone ($O_3$), an air pollutant generated through photochemical reactions
involving both anthropogenic and biogenic precursors. However, a comprehensive
evaluation of China's $O_3$ response to these multiple changes has been lacking. We
present a modelling framework under Shared Socioeconomic Pathways (SSP2-45),
incorporating future changes in local and foreign anthropogenic emissions,
meteorological conditions, and BVOCs emissions. From the 2020s to 2060s, daily
maximum 8-hour average (MDA8) $O_3$ concentration is simulated to decline by 7.7 ppb
in the warm season (April-September) and 1.1 ppb in non-warm season (October-
March) over the country, with a substantial reduction in exceedances of national $O_3$
standards. Notably, $O_3$ decreases are more pronounced in developed regions such as
BTH, YRD, and PRD during warm season, with reductions of 9.7, 14.8, and 12.5 ppb,
respectively. Conversely, in non-warm season, the MDA8 $O_3$ in BTH and YRD will
increase by 5.4 and 3.4 ppb, partly attributed to reduced $NO_x$ emissions and thereby
weakened titration effect. $O_3$ pollution will thus expand into the non-warm season in
the future. Sensitivity analyses reveal that local emission change will predominantly
influence future $O_3$ distribution and magnitude, with contributions from other factors
within $\pm 25$ %. Furthermore, the joint impact of multiple factors on $O_3$ reduction will be
larger than the sum of individual factors, due to changes in the $O_3$ formation regime.
This study highlights the necessity of region-specific emission control strategies to
mitigate potential $O_3$ increases during non-warm season and under climate penalty.

## 1 Introduction

Surface ozone ($O_3$) is a secondary air pollutant generated by photochemical
reactions in the presence of two main kinds of precursors $NO_x$ ($NO_x$=$NO$+$NO_2$) and
volatile organic compounds (VOCs). It has been reported to be a non-negligible threat
to both human health and crop yield, and also a short-lived climate forcer with



warming effect (Finlaysonpitts and Pitts, 1997; Jerrett et al., 2009; Avnery et al., 2011;
von Schneidemesser et al., 2015; Tai and Val Martin, 2017; Feng et al., 2022). Given
the abundant emissions of anthropogenic $NO_x$ and VOCs, China has suffered from
extremely high and continuously increasing $O_3$ pollution from 2013 to 2019 with the
peak season daily maximum 8-hour average (MDA8) $O_3$ concentration over 95 μg m$^{-3}$.
The rising trend has been reversed since 2020, along with the national annual $NO_x$ and
non-methane VOCs (NMVOCs) emissions reduced by 28.3 % and 3.8 %, respectively
during 2013–2020 (Zheng et al., 2018; Xiao et al., 2022; Liu et al., 2023; Wang et al.,
2023). However, current $O_3$ concentration over China is still much higher than the
global air quality guidelines (60 μg m$^{-3}$ for the averaged peak season MDA8 $O_3$, WHO,
2021). This presents a great challenge for the country to meet the criteria for public
heath welfare in the future (Feng et al., 2023; Jiang et al., 2023).

In addition to the anthropogenic driver, studies also addressed the roles of

meteorological factors, biogenic VOCs (BVOCs) emissions and transboundary
transport of pollutants on $O_3$ enhancement in China (Monks et al., 2015; Lu et al., 2019;
Cao et al., 2022; Wang et al., 2022; Weng et al., 2022; Jiang et al., 2023).
Meteorological factors, including temperature, humidity, wind, etc., influence the
chemical reactions associated with $O_3$ production and elimination, and the
transportation of $O_3$ precursors (Gong and Liao, 2019). The changed meteorology was
estimated to enhance the summer MDA8 $O_3$ concentration by 1.4 ppb yr$^{-1}$ during 2013–
2019 in the North China Plain (NCP), nearly half of the overall $O_3$ growth of 3.3 ppb
yr$^{-1}$ (Li et al., 2020a). BVOCs refer to VOCs emitted from terrestrial ecosystems and
possess high reactivity in atmospheric chemical processes, mainly including isoprene,
monoterpenes and sesquiterpene (Wu et al., 2020). Cao et al. (2022) reported that
BVOCs emissions in summer 2018 enhanced 8.6 ppb MDA8 $O_3$ averaged over China
with the highest contribution over 30 ppb in southern China. Moreover, $O_3$ and its
precursors could be transported over long distance, and transboundary foreign
anthropogenic emissions were estimated to contribute 2–11 ppb to near surface $O_3$ in



China (Ni et al., 2018; Han et al., 2019).

In the context of future global change, substantial but uncertain changes will occur

in economy, climate and land cover. According to the Sixth Assessment Report of
Intergovernmental Panel on Climate Change (IPCC AR6 report), the global surface
temperature will increase 0.2–3.7 ℃ till 2100 under different scenarios compared to
2015 (IPCC, 2021, 2022). As a result, unfavorable meteorological extremes, such as
high temperature extremes and ecological drought events, will be more frequent and
intense (Hong et al., 2019; Porter and Heald, 2019; IPCC, 2021), leading to the
deterioration of air quality named as climate penalty. To conquer the climate change
and resulting air quality deterioration, a series of measures will be implemented, such
as accelerating the transition to clean energy, upgrading industrial production
technologies and strengthening pollution control measures. Attributable to these
changes, annual mean surface $O_3$ in East Asia was projected to change by −13.9–6.1
ppb till 2100 compared to 2015 (IPCC, 2021). However, limited by the coarse
resolution of earth system model and the lack of consideration of the regional measures
for reducing air pollution and carbon emissions, global estimation is insufficient for
understanding how $O_3$ pollution in China will respond to the complex future change.

There have been some studies on how the above-mentioned changes will affect

future $O_3$ level in China. Hong et al. (2019) reported that the 1-hour maximum $O_3$ in
April to September will be enhanced by 2–8 ppb within large areas of China under
RCP4.5 (representative concentration pathways 4.5, van Vuuren et al., 2011) scenario
from the 2010s to 2050s. Under high-forcing scenarios, Li et al. (2023) projected the
climate-driven $O_3$ concentration in the 2100s and found that $O_3$ concentration in
southeast China would increase 5–20 % compared to the 2020s by a machine learning
method. A warming climate should enhance the $O_3$ level, given the increasing
frequency of atmospheric stagnation and heat waves (Hong et al., 2019; Wang et al.,
2022; Gao et al., 2023; Li et al., 2023). The effect of anthropogenic emission change
on China's $O_3$ level has been estimated by studies under different scenarios. Zhu and





Liao (2016) applied global emission estimates under RCPs, and found that the
maximum growth of annual mean $O_3$ would be 6–12 ppb during 2000–2050 under
different scenarios. Using the Dynamic Projection model for Emissions in China
(DPEC) that better includes local information of energy transition and emission
controls (Cheng et al., 2021b), Xu et al. (2022) reported that the joint impact of climate
change and emission reduction would reduce the annual MDA8 $O_3$ concentration to
63.0 μg m$^{-3}$ under ambitious scenario of carbon neutrality. Biogenic emission change
is another factor influencing future $O_3$ (Chen et al., 2009; Andersson and Engardt, 2010;
Harper and Unger, 2018; Wang et al., 2020). Liu et al. (2019) predicted a 24 % growth
of BVOCs emissions driven by climate change under RCP8.5 from 2015 to the 2050s,
resulting in a variation of daily 1-h maximum $O_3$ concentration ranging from −10.0 to
19.7 ppb across different regions in China.

Limitations exist in current studies, which prevent comprehensive assessment and

understanding of the joint impacts of future changes of multiple factors on China's $O_3$
pollution. Firstly, the above estimations mainly focused on the influence of future
changes on summertime or annual average $O_3$ concentration. As China's $O_3$ pollution
has been reported to spread into spring and fall, it is of great importance to separate the
impacts on warm (April to September, the six months with heaviest $O_3$ pollution for
most part of China, Liu et al., 2023) and non-warm season $O_3$ (October to March),
considering the diverse air pollution sources and $O_3$ formation sensitivity to precursors
for different seasons (Li et al., 2021; Wang et al., 2023). In addition, the rising
frequency of extreme weathers and declining anthropogenic emissions will further
influence the possibility of extreme $O_3$ events, which has been scarcely discussed.
Secondly, to restrain global warming, China has made a national commitment to
achieving "carbon neutrality" by 2060 (Shi et al., 2021), and accordingly launched a
series of energy and climate action plans to reduce greenhouse gas emissions. These
actions will also cause substantial reductions in air pollutant emissions, but have not
been fully included in existing predictions of global emissions (Tong et al., 2020; Cheng



et al., 2021b). Large bias will then be caused in the simulation of anthropogenic-
induced future changes of air quality, with a less realistic estimate of local emission
path (Cheng et al., 2021a). Due to probably faster decline of emissions in China but
slower in surrounding countries in the future, the contributions of transboundary
emissions on China's $O_3$ can be greatly changed and has not yet been considered.
Thirdly, BVOCs emissions will not only be affected by meteorological factors but also
by land use and land cover change (Penuelas and Staudt, 2010; Szogs et al., 2017; Wang
et al., 2021a). Future land management will change due to socio-economic development
and necessary actions as climate change response, and the changed shares of forest,
cropland and grassland will alter the magnitude and distribution of BVOCs emissions
and thereby affect $O_3$ concentration (Hurtt et al., 2020; Liao et al., 2020; Liu et al.,
2022). Finally, the existing evaluations were conducted separately for individual
influencing factors, with diverse methods and data. The interactions between different
factors were seldom included in existing analyses, and the relative contributions of
multiple factors were difficult to be evaluated or compared. Relevant studies have been
conducted in developed countries (Gonzalez-Abraham et al., 2015), and are still lack in
China.

143   In this study, we evaluate the complex influence of future changes of multiple

factors on surface $O_3$ concentration in China within a uniform framework. The
evaluation is conducted from the perspectives of seasonal, regional and extreme events
of $O_3$ pollution. Four factors are included in the analyses, i.e., meteorological conditions,
local anthropogenic emissions, BVOCs emissions, and anthropogenic emissions from
surrounding foreign countries. The analyses are conducted based on a series of
sensitivity experiments in numerical modelling of future air quality, and up-to-date
input data from multiple sources are utilized in the model (see details in next section).
We provide a comprehensive perspective on the spatiotemporal change of China's $O_3$
pollution till the 2060s, under a moderate way SSP2 of Shared Socioeconomic
Pathways (SSPs, Riahi et al., 2017) and a midrange mitigation scenario RCP4.5, a



scenario at the middle of the socio-economic developing way with radiative forcing at
4.5 W m$^{-2}$ nominally by 2100 (Meinshausen et al., 2020). The outcomes highlight the
regional and seasonal heterogeneity of O$_3$ pollution risks driven by complex future
change of multiple factors, and support strategy design of O$_3$ pollution alleviation with
specific principles, targets and action pathways.

## 2 Data and Methods

### 2.1 Main framework and research domain

The simulation framework incorporates the Weather Research and Forecasting
model (WRF, version 3.7.1) to the generate hourly meteorological fields, the Model of
Emissions of Gases and Aerosols from Nature (MEGAN, version 2.1) to calculate
gridded BVOCs emissions, and the Community Multiscale Air Quality model (CMAQ,
version 5.2) to simulate O$_3$ concentration. BVOCs emission calculation and air quality
simulations are driven by meteorological fields of 2018–2022 (the 2020s, representing
the current situation) and 2058–2062 (the 2060s, representing the future situation). All
simulation results are averaged over a period of five years to mitigate the influence of
interannual variability of meteorology. The modelling domain, same for WRF,
MEGAN and CMAQ, covers East Asia, most areas of South Asia and Central Asia, and
part of Southeast Asia and North Asia (Figure 1). It applies the Lambert Conformal
Conic projection centered at (110° E, 34° N), and the horizontal resolution is 27 km×27
km, with 303×203 grids. The target area, Chinese mainland, includes 31 provincial-
level administrative regions (excluding Hong Kong, Macao and Taiwan). Eight
geographical regions are defined, and locations of the three regions with dense
population and relatively heavy air pollution are also shown in Figure 1, namely BTH
(Beijing-Tianjin-Hebei), YRD (Yangtze River Delta) and PRD (Pearl River Delta).

### 2.2 Data sources and processing methods

We use the bias-corrected RCP4.5 output of the National Center for Atmospheric



Research's Community Earth System Model (NCAR CESM) as initial and boundary
conditions for WRF (Monaghan et al., 2014). A ten-year dynamic downscaling
simulation for 2018–2022 and 2058–2062 is conducted. Note we do not utilize the real-
time reanalysis data to drive the simulation of the 2020s, in order to minimize the
systematic error between the simulation driven by real meteorological conditions (for
current simulations) and climate projection (for future simulations),
The BVOCs emissions are basically determined by meteorology and vegetation.
The meteorological conditions are supplied by WRF. The vegetation data, including
leaf area index (LAI), plant functional types (PFTs) and emission factors (EFs) of each
PFT, are determined for 2020 and 2060. Gridded LAI data for 2020 are obtained from
Global Land Surface Satellite product (Liang et al., 2021), and those for 2060 under
SSP2-45 scenario are downscaled from the daily CESM2 output of Coupled Model
Intercomparison Project Phase 6 (CMIP6). PFTs data for 2020 are derived from
MCD12C1 product of Moderate-Resolution Imaging Spectroradiometer (MODIS)
dataset and mapped to the 16 types required for MEGAN following Liao et al. (2020).
The PFTs data for 2060 in China are obtained from Liao et al. (2020) under SSP2-45
scenario, with other regions maintaining those of 2020. EFs for each PFT are taken
from Guenther et al. (2012).
Anthropogenic emissions for Chinese mainland are obtained from the Multi-
resolution        Emission        Inventory        for        China        (MEIC,
http://meicmodel.org.cn/?page_id=560) for 2020, and DPEC version 1.1 under SSP2-
45 incorporating the best available end-of-pipe pollution control technologies for 2060.
Emissions outside Chinese mainland are obtained from CMIP6 dataset under SSP2-45
scenario (O'neill et al., 2016; Gidden et al., 2019). The spatial and temporal
distributions of emissions outside Chinese mainland are assumed the same as those in
MIX Asian emission inventory (Li et al., 2017). The speciation profiles of NMVOCs
are taken from MIX as well. Supplementary Figure S1 shows the emissions of two main
precursors of $O_3$ by year and region. The $NO_x$ and NMVOCs emissions for Chinese





mainland were estimated to decline 58 % and 51 % from 2020 to 2060, respectively,
much faster than those of surrounding areas within the modelling domain (8 % and 14 %
respectively). In particular, the NO$_x$ emissions would decline 57–62 % for the three
developed regions BTH, YRD and PRD, while the reductions of anthropogenic
NMVOCs would vary a lot among regions (36 %, 49 % and 60 % for BTH, YRD and
PRD, respectively).

Carbon Bond 2005 (CB05, Yarwood et al., 2005) is adopted as the gas-phase

chemical mechanism and the sixth-generation CMAQ aerosol module AERO6 (Appel
et al., 2013) as aerosol chemistry mechanism. The initial and boundary conditions are
set by default clean air conditions in CMAQ, and the first 10 days for each year are
determined as the spin-up period to minimize the effects of initial and boundary
conditions.

**2.3 Simulation cases**

Six cases of CMAQ simulations are conducted to investigate the impacts of future

change of the four factors on O$_3$ concentration in China (Table 1). Cases 1 and 2
represent the current (2020s) and future (2060s) baseline, respectively, and the
difference between them indicates the joint effect of the future changes of multiple
factors. Each of Cases 3–6 applies the prediction for 2060s for one specific factor but
keeps the remaining factors at current condition (2020s). Thus, the difference between
each of those four cases and Case 1 indicates the impact of individual factor, including
meteorological conditions (Case 3), domestic anthropogenic emissions (Case 4),
BVOCs emissions (Case 5) and anthropogenic emissions of surrounding countries
(Case 6). Each case contains a five-year (2018–2022 or 2058–2062) WRF-MEGAN-
CMAQ simulation driven by the varying meteorological conditions for individual years,
and the five-year average of simulated O$_3$ concentrations is adopted for further analyses.

**2.4 Model performance**

To evaluate the model performance, we conduct a comparative analysis between



simulations and observations for meteorological factors and $O_3$ concentrations, as well
as an intercomparison for BVOCs estimates between different studies.
We first examine the capability of downscaled CESM climate projections in
capturing the meteorological conditions of the 2020s. We applied the meteorological
data from the National Climate Data Center (NCDC, archived at https://quotsoft.net/air)
in 2020, and the statistical metrics are presented in Supplementary Table S1. The
modeled temperature at 2 m (T2) is in good spatiotemporal agreement with the
observations, with the correlation coefficient (R) of 0.96 and index of agreement (IOA)
of 0.98. The relative humidity (RH) is also well predicted with R and IOA at 0.78 and
0.88, respectively. The model shows an overestimation on the wind speed by 1.41 m
$s^{-1}$, which is also reported by Hu et al., (2022). The correlation coefficients of wind
speed and direction are higher than 0.5. Overall, the modeled meteorological fields have
basically captured the conditions in China and are appropriate for subsequent MEGAN
and CMAQ simulations.
For BVOCs emissions, we compare our estimates for the 2020s with previous
studies, as summarized in Supplementary Table S2. The total BVOCs, isoprene and
terpenes emissions in this study are estimated at 33.55, 21.08 and 3.30 Tg $yr^{-1}$,
respectively, and are comparable to other studies. In particular, our estimate is larger
than others except for Li et al. (2020b) for isoprene, while smaller than others except
for Wu et al. (2020) for terpenes. The differences between studies might result from the
diverse strategies of mapping PFTs from the original satellite products and the
difference between downscaled climate conditions and the real meteorological fields.
We apply the observed MDA8 $O_3$ concentration data from the national network of
China Ministry of Ecology and Environment (archived at https://quotsoft.net/air) to
evaluate CMAQ performance. As shown in Supplementary Figure S2, the simulation
could capture the spatiotemporal distribution of surface MDA8 $O_3$ concentration for the
whole country and specific $O_3$ pollution hot spots, e.g., BTH and eastern Sichuan
province with their surrounding areas. The statistical metrics of the comparisons



between the simulated and observed monthly average MDA8 $O_3$ concentration of 2020s
are summarize in Supplementary Table S1. The normalized mean biases (NMB) are
calculated at 14.12 % and 10.90 % for warm and non-warm season, and R values at
0.71 and 0.32, respectively. Even with a slight overestimation, the reliability of our
simulation is comparable to most previous studies in China, with a better performance
in the warm season (Hu et al., 2016; Lu et al., 2019; Gao et al., 2020; Yang and Zhao,

2023).

## 3 Results and Discussions

### 3.1 Future change of meteorology and BVOCs emissions

The downscaled changes in the meteorological factors from the 2020s to 2060s
(SSP2-45 scenario) are shown in Figure 2, including temperature, RH and wind speed
(WS). The changes are analyzed separately for April–September (warm season) and
October–March (non-warm season). For the warm season, daily maximum temperature
at 2 m (T-max) will increase across China with an average change of 1.0 ℃, and the
minimum and maximum changes are found in Tibetan Plateau at 0.1 ℃ and in
Heilongjiang province at 2.1 ℃, respectively. The RH will decrease slightly by −0.6 %
for the whole country, with the changes for most areas within the range between −3 %
and 0 % except for some areas of Northwestern China, Southwestern China, and
Tibetan Plateau (see the region definitions in Figure 1). The growing T-max and
declining RH will enhance the photochemical production of $O_3$ and BVOCs emissions.
For the non-warm season, the national average growth of T-max will be smaller at 0.2 ℃
and some areas in Northeastern, Northern and Eastern China will even experience a
decline ranging from −1.8 to 0 ℃. The RH will change diversely across the country,
ranging from −6.0 to 6.3 %. Very limited change in WS will occur, ranging from −0.1
to 0.2 m s$^{-1}$ in most areas of the country. The spatial distribution of downscaled future
meteorological field changes is generally in agreement with those predicted by Hong et
al. (2019) and Hu et al. (2022). Some discrepancies in temperature and wind speed





change of non-warm season between studies result from the different choices of base
year and parameterization schemes of WRF.

Table 2 shows China's BVOCs emissions of the 2020s and 2060s (SSP2-45

scenario) estimated with MEGAN, as well as the BVOCs emission intensity (emissions
per unit area) for the three developed regions. The emissions will increase from 33.6
Tg yr$^{-1}$ for the 2020s to 43.4 Tg yr$^{-1}$ for the 2060s. The growth rates in BTH, YRD and
PRD are predicted to be 22.4 %, 23.9 % and 23.0 %, respectively, smaller than that for
the whole country (29.2 %). The spatial distributions of BVOCs emissions for the 2020s
and the changes from the 2020s to 2060s, are shown in Supplementary Figure S3. Areas
all over China will experience the growth of BVOCs emissions, and it will be more
prominent in areas with high vegetation coverage (e.g., Southern and Southwestern
China) rather than urban areas. The growth of BVOCs emissions will enhance the
contribution of natural sources to $O_3$ formation, especially along with declining
anthropogenic emissions in the future (Penuelas and Llusia, 2003; Riahi et al., 2017;
Gao et al., 2022).
**3.2 Response of surface $O_3$ concentration to combined future changes**

Figure 3 illustrates the spatial distributions of MDA8 $O_3$ concentrations for the

warm and non-warm seasons of the 2020s and 2060s (SSP2-45 scenario), as well as the
differences between the two periods. Briefly, future changes of the four factors under
SSP2-45 are estimated to jointly reduce MDA8 $O_3$ by 7.7 and 1.1 ppb in the warm and
non-warm season, respectively, while the $O_3$ responses to future changes will differ by
region.

In the warm season of the 2020s (Figure 3a), the nationwide average MDA8 $O_3$

concentration is simulated at 57.3 ppb, and those of BTH, YRD and PRD are 73.7, 68.7
and 52.3 ppb, respectively. Hot spots of $O_3$ pollution, with average MDA8 $O_3$ over 75
ppb, are mainly located in Northern China and Sichuan province. The pattern is
predicted to persist into the 2060s (Figure 3b), with a decline in both the severity and
size of highly polluted regions. The nationwide MDA8 $O_3$ concentration will decline



13.4 % to 49.6 ppb, and that in most areas of China will be within the range of 37.5–
67.5 ppb. The highest concentration will be lower than 75 ppb for the two hotspots of
Northern China and Sichuan. BTH will remain as the most $O_3$-polluted area in warm
season, with the $O_3$ concentration at 63.9 ppb (13.3 % smaller than the 2020s), while
that of YRD and PRD will decrease to 53.9 (21.5 %) and 39.8 ppb (23.9 %),
respectively. $O_3$ concentration in the developed regions will decline faster than or
roughly the same as that for the whole country. The reductions in MDA8 $O_3$ from 2020s
to 2060s will be 10–20 ppb for Northern, Eastern, Central and Southern China and 0–
10 ppb for Northeastern and Northwestern China as well as the Tibetan Plateau (Figure
3c). Notably, some areas in Sichuan are expected to experience a substantial decline of
MDA8 $O_3$ over 20 ppb.

$O_3$ concentration of the non-warm season is simulated to be much lower than that

of the warm season. The 2020s average MDA8 $O_3$ is 48.4 ppb, ranging from 30.0 to
67.5 ppb in most areas of China (Figure 3d). Different from the warm season in which
highest concentration is found for Northern China and Sichuan, the Southern and
Southwestern parts of China suffer the highest $O_3$ level for the non-warm season. A
general west-to-east and south-to-north gradient is found for MDA8 $O_3$, with the lowest
concentration found in Northern and Northeastern China. The concentrations in BTH
and YRD are simulated at 33.8 and 45.1 ppb, respectively, much lower than that of
PRD (58.9 ppb). Relatively high temperature during even the non-warm season is
expected to expand the $O_3$ pollution period in Southern China. Resulting from complex
change of multiple factors, the national average MDA8 $O_3$ concentration in the non-
warm season of 2060s will decrease slightly to 47.3 ppb under SSP2-45, and that in
most regions will be within the range of 37.5–52.5 ppb except for some areas in
Northeastern China and Tibetan Plateau (Figure 3e). The MDA8 $O_3$ concentrations of
the three developed regions will become closer at 39.3, 48.4 and 51.6 ppb for BTH,
YRD and PRD, respectively. As illustrated in Figure 3f, MDA8 $O_3$ is predicted to
increase in BTH and YRD and the surrounding areas, with the growth mostly ranging



0–15 ppb. In other areas (especially in Southern China), the concentration will decrease
in the non-warm season by −15 to −5 ppb. As a result of the increased $O_3$ in the less
polluted Eastern and Northern China and decreased $O_3$ in the more polluted
Southwestern and Southern parts, the 2060s regional disparity in the non-warm season
$O_3$ pollution will get smaller compared to the 2020s (Figure 3d and 3e).

To further explore the temporal pattern of $O_3$ level in the future, we compare the

monthly average MDA8 $O_3$ in the 2020s and 2060s under SSP2-45 for the whole
country and three developed regions (Figure 4 and Supplementary Figure S4). For the
whole country (Figure 4a), the changes of monthly average MDA8 $O_3$ from 2020s to
2060s are estimated to range from −3.2 to −10.7 ppb in the warm season but less
prominent in the non-warm season (from −2.7 to 0.9 ppb). Along with the more
reduction in summertime (June, July and August), in particular, the periods with the
highest $O_3$ concentration will expand into spring (March) and fall (October), as
presented in Supplementary Figure S4. For the three regions, a greater decline in $O_3$
concentration is found in the warm season while a smaller or even a growth is found in
the non-warm season. For BTH (Figure 4b), the monthly MDA8 $O_3$ concentrations
range between 24.7 and 88.4 ppb in the 2020s with a clear difference between the warm
and non-warm season. This pattern will remain in the 2060s with smaller difference
between months (30.6–70.2 ppb). The temporal change pattern of YRD is similar to
that in BTH, with decline in the warm season and growth in the non-warm season
(Figure 4c). The shift of $O_3$ pollution from the warm towards the non-warm season is
more prominent in the PRD, the only region where $O_3$ concentration of all the months
in 2060s is predicted to decline (Figure 4d). Different from BTH and YRD, as
mentioned above, higher $O_3$ concentrations during spring and autumn and lower in
summer (due to the abundant summertime precipitation and high humidity) are found
for PRD in the 2020s (Gao et al., 2020; Han et al., 2020). With great $O_3$ decline in the
warm season, the periods experiencing peak $O_3$ pollution are predicted in the non-warm
season of the 2060s, predominantly between October and March (Supplementary



Figure S4).

### 3.3 Identifying surface O₃ response to individual factors


### 3.3.1 Local anthropogenic emission change


Figure 5 shows the influences of changes of each individual factors (local
anthropogenic emissions, meteorological conditions, BVOCs emissions, and
anthropogenic emissions from surrounding countries) on the warm and non-warm
season O₃ concentrations. Out of the four, the change of local anthropogenic emissions
is predicted to be the most influential factor, resulting in a national average decline of
7.2 and 0.8 ppb for the warm and non-warm season, respectively (Figure 5a and 5e). In
the warm season, the emission reduction will play a positive role in reducing O₃
pollution in most areas of China, and the decrease will exceed 10 ppb across Northern,
Eastern, Central, Southern and part of Southwestern China. In the non-warm season,
emission reduction will have contrasting effects on MDA8 O₃ levels in the north and
south part of China, enhancing MDA8 O₃ by 0–15 ppb for the former while restraining
it by 0–10 ppb for the latter. Especially, the emission reduction is predicted to elevate
the O₃ concentration by 5.9 and 4.0 ppb for BTH and YRD respectively.
Supplementary Figure S5 shows the relative emission reductions from 2020s to
2060s by region. Under SSP2-45 scenario, the reductions of $NO_x$ and VOCs emissions
will range from 35.6 % to 63.6 % for different regions, and VOCs emission reduction
will be less than that of $NO_x$ except for PRD. As the $NO_x$-limited regime for O₃
formation (i.e., O₃ is more sensitive to $NO_x$ emission change) occurs more frequently
in the warm season while the VOC-limited regime more in the non-warm season, the
larger decline of $NO_x$ emissions than VOCs should be more effective in restraining the
warm season O₃ pollution but has less benefit or even negative effect in the non-warm
season (Sillman and He, 2002). Wintertime of NCP and YRD have been reported under
the VOC-limited regime and the excessive $NO_x$ emissions play an important role in
removing O₃ by titration effect (Jin and Holloway, 2015; Li et al., 2021; Wang et al.,



2021b). This may explain the MDA8 $O_3$ increase during the non-warm season with
insufficient reduction of VOCs (35.6 % and 49.5 %) but sharp reduction of $NO_x$ of 53.4 %
and 60.3 % for NCP and YRD, respectively. Supplementary Figure S6 shows the
monthly variation of $O_3$ and odd oxygen ($O_x$, $O_x=O_3+NO_2$, representing the real
photochemical production potential of $O_3$ considering the titration effect) in the 2020s
and 2060s. It should be noted that the growth of $O_3$ in the non-warm season in 2060s
for BTH and YRD will be accompanied by minimal change of $O_x$, while the declines
of $O_3$ and $O_x$ will appear simultaneously in the warm season for the three regions and
in non-warm season for PRD. This indicates that the growth of non-warm season $O_3$ in
BTH and YRD should result partly from $NO_x$ reduction and thereby weakened NO
titration, as titration is a key pathway of $O_3$ loss when the chemical reactivity is
relatively low in winter (Gao et al., 2013; Akimoto and Tanimoto, 2022). The
differentiated $O_3$ responses to precursor reduction between YRD and PRD have also
been detected during the COVID-19 breakout period. With the $O_3$ isopleth plots, Wang
et al. (2021b) illustrated that 40–60 % reduction of $NO_x$ and VOCs enhanced the $O_3$
formation in YRD under the VOC-limited regime but suppressed $O_3$ in PRD under the
transitional regime (a regime between $NO_x$- and VOC-limited). Therefore, VOCs
emission controls should be better addressed for $O_3$ pollution alleviation when it
expands to non-warm season in the future.
**3.3.2 Meteorological condition change**

As shown in Figure 5b and 5f, the influence of meteorological change exhibits

different patterns for the warm and non-warm season.

In the warm season, meteorological change due to global warming will play a

positive role on $O_3$ formation in most of China, with the enhancement within 0–4 ppb,
but it will reduce the $O_3$ level in remote areas like Tibetan Plateau. The national average
growth will be 0.3 ppb and that for YRD, PRD and BTH will be 1.9, 0.7, and 0.3 ppb,
respectively. The response of $O_3$ to meteorological change is associated with some
specific variables (Hong et al., 2019). For example, the great enhancement of $O_3$ in




YRD might be attributable to a hotter, dryer and more stable atmosphere with growth
of T-max (over 0.6 ℃) and decline of RH and WS (Figure 2). The result is similar to
Hong et al. (2019), which reported a change of 2–8 ppb of daily 1-hour maximum $O_3$
concentration for the peak season from the 2010s to 2050s under RCP4.5. In addition,
the declining $O_3$ in Tibetan Plateau and the surrounding areas might result partly from
the weakened long-range transport of peroxyacetyl nitrate (PAN, the principal $NO_x$
reservoir) from the polluted areas (Fischer et al., 2014). Driven by the elevated
temperature, PAN from relatively polluted regions will undergo stronger thermal
decomposition locally, thus fail to be transported far away to the remote regions to
promote $O_3$ formation (Liu et al., 2013; Lu et al., 2019).
The influence of meteorological change on $O_3$ production is predicted to be much
smaller for the non-warm season, with the magnitude within ±1 ppb in most areas and
nationwide average at −0.2 ppb. In the three developed regions, the changes are
predicted to range from −0.4 to 0.3 ppb, with little regional difference. The limited
influence might be attributable to the modest change in temperature and RH in the non-
warm season.
**3.3.3 BVOCs and surrounding anthropogenic emission change**
Compared to domestic emissions, change of BVOCs emissions and anthropogenic
emissions from surrounding countries will have a less influence (within ±3 ppb) on
surface $O_3$ concentration in China. BVOCs change tends to enhance $O_3$ while foreign
emission change tends to restrain it in most areas (Figure 7).
The growing BVOCs emissions due to vegetation and climate change is estimated
to enhance $O_3$ concentration by 0–3 ppb in the most areas of China, with a larger
influence of 0.6 ppb in the warm season than that of 0.3 ppb in the non-warm season
across the country (Figure 5c and 5g). In the warm season, relatively large growth of
$O_3$ concentration will occur in BTH at 2.1 ppb, and those of YRD and PRD will be 1.5
and 1.0 ppb, respectively. The abundant $NO_x$ emissions in BTH are expected to result
in a larger $O_3$ concentration response to BVOCs emission change than YRD and PRD,





even the BVOCs emission change of BTH will be smaller than the other two regions
(Table 2). The result is in agreement with other numerical simulation experiments. Liu
et al. (2019) reported a prominent $O_3$ enhancement even with a low BVOCs emission
rate under RCP8.5, in a $NO_x$-abundant environment. In the remote areas like Tibetan
Plateau and part of Northeastern China, the increased BVOCs will remove $O_3$ due to
the isoprene ozonolysis in low-$NO_x$ environment (Hollaway et al., 2017; Zhu et al.,
2022). In general, regions with higher $O_3$ pollution levels and $NO_x$ emissions will suffer
more risk of $O_3$ growing from rising BVOCs emissions in the future.
Most areas of China will benefit from the foreign emission change in terms of $O_3$
pollution alleviation (Figure 5d and 5h). An exception is Tibetan Plateau and its
surrounding areas, which will be affected by the elevated emissions of $NO_x$ and VOCs
from South Asia under SSP2-45. Limited by the range of pollutant transport, greater
impacts will be found for coastal and border areas and less for inland areas (Ni et al.,
2018). Larger $O_3$ changes in the three developed regions are predicted than that of the
whole country, benefitting from the precursor emission reduction in East Asia and
Southeast Asia.
**3.4 The relationship between the joint and separate effects of multiple factors**
Figure 6 summarizes the contributions of individual factors to the total $O_3$ change
by region and season. Due to the nonlinear response of $O_3$ to multi-factor changes, the
aggregated contribution of the four factors does not equal to the joint contribution (i.e.,
there exist gaps between the difference of the 2020s and 2060s and the aggregated
contribution of four factors).
The varying domestic anthropogenic emissions are predicted to dominate the
change of the future $O_3$, with a relative contribution ranging from 75 % to 117 % for
different regions and seasons. The relative contributions of the other three factors are
estimated to be limited within ±25 % at national and regional level. Among different
regions, YRD will be more affected by climate change with the contribution of −13 %
and −12 % for the warm and non-warm season, respectively, far greater than that of





BTH and PRD (−6 % to 0 %). BTH will be more affected by BVOCs emission change
than other regions in the warm season (−21 %), while YRD and PRD will be more
affected in the non-warm season with the relative contributions of 17 % and −20 %,
respectively. Little regional difference is found for the relative contributions of foreign
emission change.

To better understand the regional and seasonal differences of the relative

contributions of future changes to $O_3$ concentration, we examine the nonlinear response
of $O_3$ to precursor change in the three developed regions. We follow Chen et al. (2021)
and Schroeder et al. (2017), and conduct a fit of lognormal distribution for the
relationship of modeled hourly $O_3$ and $NO_2$ concentrations, as shown in Figure 7. The
data points on the left of the turning point of fitted curve suggest a $NO_x$-limited regime
while on the right a VOC-limited regime, and data points around the turning point are
under transitional regime.

The $O_3$-$NO_2$ relationship from the 2020s to 2060s will be mostly influenced by the

changing domestic anthropogenic emissions, indicated by the close distributions of data
points and fitted curves between "EMIS" and "2060s" in Figure 7. In the warm season,
the future $O_3$-$NO_2$ relations in BTH and YRD are predicted to change greatly from a
highly $O_3$ polluted situation with moderate $NO_2$ concentration to a situation with a
relatively low level of $NO_2$ (mostly under 10 ppb) and a moderate level of $O_3$ (under
60 ppb). A weak VOC-limited regime appeared for the whole BTH in 2020s, and there
is big diversity within the region, including a dense area with strong VOC-limited
regime and other areas with transitional or $NO_x$-limited regime (Figure 7a). Represented
by the moving of most points from the right of the turning point to near or left of the
turning point, the $NO_x$-limited and transitional regimes will dominate BTH in the 2060s.
Compared to 2020s, the data points of 2060s are more closely distributed, indicating a
reduced diversity of $O_3$ formation regime in the region. For YRD, most areas were
under transitional or weak VOC-limited regime in the 2020s with limited diversity
within the region, and the situation in 2060s will be similar to that of BTH (Figure 7a



and 7b). The shift from weak VOC-limited regime in 2020s to transitional or $NO_x$-
limited regime in 2060s for BTH and YRD implies the influence of emission reduction
on altering the sensitivity of $O_3$ formation to precursors. Most areas of PRD in the 2020s
are under transitional or $NO_x$-limited regimes, and the regime will transfer to a strong
$NO_x$-limited one in 2060s, with an almost positive correlation between $NO_2$ and $O_3$ in
a low-$NO_2$ environment (Figure 7c). In the non-warm season, $O_3$ and $NO_2$ will remain
negatively correlated for BTH and YRD till the 2060s, which suggests a persistent
VOC-limited regime and explains the $O_3$ concentration growth along with substantial
precursor emission reductions. The turning points are simulated at extremely low $NO_2$
concentrations of 2.0 and 1.2 ppb for BTH and YRD, respectively (Figure 7d and 7e).
A big challenge still exists on effective emission controls to reduce the $O_3$ concentration
in the non-warm season for the two regions. Differently, the $O_3$ formation sensitivity in
most of PRD will shift from transitional regime towards a more $NO_x$-limited situation
(Figure 7f).

The fitted curves of other three factors are similar to those of the 2020s, and the

change of these factors will make little difference on $NO_2$ concentration but will result
in moderate changes on $O_3$ concentration within $\pm 2$ ppb. The limited changes of climate,
BVOCs emissions and foreign anthropogenic emissions will not essentially alter the $O_3$
formation regime, but may change the $O_3$ production under the nearly same $NO_2$
concentration. Changes of individual meteorological factors are expected to easily
influence the $O_3$ and $NO_2$ concentrations (Pope et al., 2015; Liu and Wang, 2020;
Dewan and Lakhani, 2022). The modeled little response of $NO_2$ to meteorological
change, except that in the non-warm season for BTH, might be attributed to the
compensating effect of different variables. The limited influence of BVOCs on the $O_3$
formation sensitivity to precursors is consistent with Gao et al. (2022), which reported
comparable empirical kinetic modelling approach (EKMA) curves with and without
BVOCs emissions. The transboundary $O_3$ pollution results from the transport of both
$O_3$ and its precursors (mainly associated with PAN), while $NO_2$ is less influenced by



long-range transport due to its shorter lifetime (Ni et al., 2018; Yin et al., 2022).

The change in $O_3$ formation regime might partly explain the finding that the joint

effect of multiple factors on restraining $O_3$ pollution will be larger than the aggregated
effects of individual factors. Under a $NO_x$-limited regime, $O_3$ is less sensitive to
changing VOCs emissions (e.g., BVOCs emissions) than that under a VOC-limited one.
Therefore, the enhancement of $O_3$ due to BVOCs emission growth in the future will be
restrained with a much lower $NO_2$ concentration. This indicates a co-benefit of reducing
the anthropogenic emissions to restrain the potential $O_3$ pollution elevation due to
growing BVOCs emissions (as a part of climate penalty) in the future.
**3.5 Change of $O_3$ exceedance events over the east of China**

Figure 8 shows the "$O_3$ exceedance events" over the east of China (mainly

including Northern, Eastern, Central and Southern China) in the 2020s and 2060s, and
the changes influenced by different factors. The exceedance is defined as number of
days with the MDA8 $O_3$ exceeding the Chinese National Air Quality Standard-Grade
II (160 μg m$^{-3}$ or 81.6 ppb). The exceedance events appear mainly in the warm season
(Figure S7). Areas with frequent exceedance (over 50 days) in the 2020s were mainly
located in Northern China. Much fewer exceedances are found for YRD and PRD (19.3
and 8.2 days in 2020s, respectively). In the 2060s, the $O_3$ exceedance events will drop
significantly. The exceedance days will be fewer than 10 days for most of the country,
except for some areas in BTH which will still have more than 20 exceedance days over
the year.

Domestic emission abatement will be the most important factor reducing the $O_3$

exceedance, particularly in Northern China. The exceedance days will be cut by 45.3,
19.1 and 8.1 days for BTH, YRD and PRD, respectively, with the maximum reduction
reaching 80 days within BTH and YRD. Notably, the spatial pattern of changing $O_3$
exceedance due to emission reduction is different from that of changing MDA8 $O_3$ due
to emission reduction as shown in Figure 5a. Even the warm season MDA8 $O_3$
concentration of BTH will decline only 9.7 ppb, the $O_3$ exceedance events will be



greatly reduced, indicating that national emission controls will be especially effective
in reducing serious $O_3$ pollution. Climate change will mainly affect Jiangsu, Anhui,
Henan and Hebei provinces, elevating the exceedance by more than 15 days in most of
these areas. For YRD and PRD, climate change will elevate the exceedance by 9.5 and
3.3 days, respectively. Some areas of BTH will benefit from climate change, with the
exceedance declining 0–10 days. The influences of BVOCs and foreign emission
change on exceedance days are of limited regional differences, with a growth of 5 to 15
days for the former and a decline of −5 to 0 days for the latter. The exceedances elevated
by BVOCs emission growth will be 6.6, 6.1 and 2.8 days for BTH, YRD and PRD with
the maximum reaching 19, 18 and 12 days within the region, respectively, reflecting an
unneglectable role of biogenic source change on future $O_3$ episodes.

## 4 Conclusions

We explore the response of China's surface $O_3$ concentration to the future changes
of multiple factors under SSP2-45, based on a series of sensitivity experiments with
WRF-MEGAN-CMAQ simulations. From the 2020s to 2060s, the MDA8 $O_3$
concentration is predicted to decline by 7.7 and 1.1 ppb in the warm and non-warm
season, respectively, and the $O_3$ exceedances of Chinese National Air Quality Standard
(Grade II) will be largely eliminated. In the warm season, MDA8 $O_3$ in BTH, YRD and
PRD will decline by 9.7, 14.8 and 12.8 ppb, respectively, larger than the national
average level. However, MDA8 $O_3$ will increase in BTH and YRD in the non-warm
season attributed to the reduced $NO_x$ emissions and thereby titration effect. The $O_3$
pollution will expand towards the non-warm season in the future, bringing new
challenge for policy makers to optimize the strategy of precursor emission controls
based on local conditions.
Reduction of local anthropogenic emissions is estimated to dominate the spatial
distribution and magnitude of future $O_3$ change. Meteorological variation will lead to a
change of MDA8 $O_3$ ranging between −1 and 4 ppb for most areas in the warm season.



The influences of changing BVOCs and foreign anthropogenic emissions will be within
±3 ppb, with the former elevating $O_3$ while the latter reducing $O_3$. Especially in areas
with high $O_3$ pollution and intense $NO_x$ emissions, the growing BVOCs emissions will
more enhance the risk of $O_3$ pollution. The joint effect of multiple factors on restraining
$O_3$ pollution will be larger than the aggregated effects of individual factors, which can
be partly explained by the changing $O_3$ formation regime. Large amount of emission
reduction under SSP2-45 will reshape the $O_3$ formation sensitivity to precursors. In
BTH and YRD, $O_3$ formation in the warm season is projected to shift from weak VOC-
limited to transitional or $NO_x$-limited regime, while VOC-limited regime will still
dominate in the non-warm season. In the future, $O_3$ will be less sensitive to BVOCs
change in a low $NO_x$ environment along with persistent emission controls, highlighting
the benefit of anthropogenic emissions abatement on mitigating the climate penalty and
limiting $O_3$ pollution.
Limitations exist in current study. Firstly, the future climate data are taken from
one single model CESM, subject to bias in the assessment of meteorological influence
on $O_3$. Secondly, some factors that will influence future $O_3$ level are not included in our
analyses, such as the changing $CH_4$ concentration and the stratosphere-troposphere
exchange of $O_3$. Thirdly, there exist gaps between the downscaled and realistic
conditions of meteorology for the 2020s, leading to uncertainty in the $O_3$ simulation.
Finally, the changing $O_3$ formation regime is presented through the relation between $O_3$
and $NO_2$ concentrations, and the mechanism how the climate penalty will influence $O_3$
formation under substantial reduction of anthropogenic emissions needs to be better
analyzed in future studies.

## Data availability

All data in this study are available from the authors upon request.

## Author contributions

JYang developed the methodology, conducted the work and wrote the draft. YZhao



improved the methodology, supervised the work and revised the manuscript. YWang
and LZhang contributed to the methodology and provided supports to the scientific
interpretation and discussions.
**Competing interests**
The authors declare that they have no conflict of interest.
**Acknowledgments**
This work was sponsored by the National Key Research and Development
Program of China (2023YFC3709802), National Natural Science Foundation of China
(42177080), and the Key Research and Development Programme of Jiangsu Province
(BE2022838). We thank Qiang Zhang and Dan Tong from Tsinghua University for the
emission data (MEIC and DPEC).



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





**FIGURE CAPTIONS**
Figure 1 The modelling domain and geographical definitions (denoted by colors) of this
study. Boundaries of the three regions, including BTH (Beijing-Tianjin-Hebei), YRD
(Yangtze River Delta) and PRD (Pearl River Delta), are marked by dark grey lines.
Figure 2 Projected changes of the strongly ozone-related meteorological elements, daily
maximum temperature at 2 m (T-max, a and d), relative humidity (RH, b and e) and
wind speed (WS, c and f), from the 2020s to 2060s. Panels (a-c) represent those of the
warm season, and panels (d-f) represent those of non-warm season.
Figure 3 Simulation and projection of seasonal average MDA8 $O_3$ in the 2020s (Case1,
a and b) and 2060s (Case2, d and e), and the changes over this period (Case2−Case1, c
and f). Panels (a-c) represent those of the warm season, and panels (d-f) represent those
of non-warm season. Regional mean concentrations across China (CHN), BTH, YRD
and PRD are inset.
Figure 4 Simulation and projection of monthly average MDA8 $O_3$ in the 2020s and
2060s across CHN (a), BTH (b), YRD (c) and PRD (d).
Figure 5 Projected changes of MDA8 $O_3$ from the 2020s to 2060s attributed to
anthropogenic emissions from local sources (Case3−Case1, a and e), meteorological
conditions (Case4−Case1, b and f), BVOCs emissions (Case5−Case1, c and g) and
anthropogenic emissions from surrounding countries (Case6−Case1, d and h). Panels
(a-d) represent those of the warm season, and panels (e-h) represent those of non-warm
season. Regional mean changes across CHN, BTH, YRD and PRD are inset.
Figure 6 The relationships between the separate MDA8 $O_3$ changes attributed to the
four factors (denoted by the name of Case3–6) and the total changes from the 2020s to
2060s over China and the three regions. Panels (a-d) represent those of the warm season,
and panels (e-h) represent those of non-warm season. The relative contributions of the



four factors to the total influence of future change are shown in the light grey box.
Figure 7 The relationships between simulated hourly $NO_2$ and $O_3$ concentrations with
the lognormal fits for different regions and seasons. The colored circles, representing
different cases, come from the seasonal average concentrations for each grid in the
target region. The specific circles with black border represent the regional average
situation, and the turning points of every fitted curve are marked by the "+" sign. The
density plots of the 2020s and 2060s are inset.
Figure 8 Projected annual $O_3$ exceedance over the east of China in the 2020s and 2060s,
and the exceedance changes when the four factors at 2060s level. Regional mean
changes across CHN, BTH, YRD and PRD are inset.



**TABLES**

**Table 1. List of simulation cases to investigate the impact of future change upon surface O₃ in China, with sensitivity experiments from the perspectives of four main influencing factors.**

| Case number | Case name | China's local emissions | Meteorological conditions | BVOCs emissions | Surrounding emissions |
|---|---|---|---|---|---|
| Case1 | 2020s | 2020 | 2018-2022 | 2018-2022 | 2020 |
| Case2 | 2060s | 2060 | 2058-2062 | 2058-2062 | 2060 |
| Case3 | EMIS | 2060 | 2018-2022 | 2018-2022 | 2020 |
| Case4 | CLIM | 2020 | 2058-2062 | 2018-2022 | 2020 |
| Case5 | BVOC | 2020 | 2018-2022 | 2058-2062 | 2020 |
| Case6 | SURR | 2020 | 2018-2022 | 2018-2022 | 2060 |

**Table 2. The BVOCs estimation over China and emission intensity in BTH, YRD and PRD of the 2020s and 2060s, as well as the corresponding growth rates over this period.**

| | China | BTH | YRD | PRD |
|---|---|---|---|---|
| | Emissions (Tg) | Emission intensity (Gg grid$^{-1}$) | | |
| 2020s | 33.6 | 1.4 | 4.6 | 8.7 |
| 2060s | 43.4 | 1.7 | 5.7 | 10.7 |
| Growth rate | 29.2 % | 21.4 % | 23.9 % | 23.0 % |



**FIGURES**

**Figure 1**

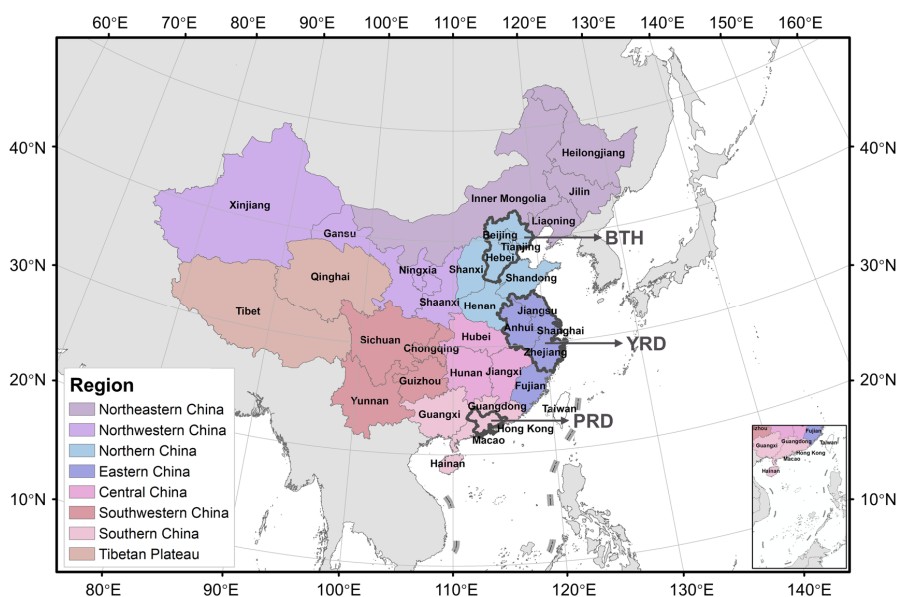



**Figure 2**

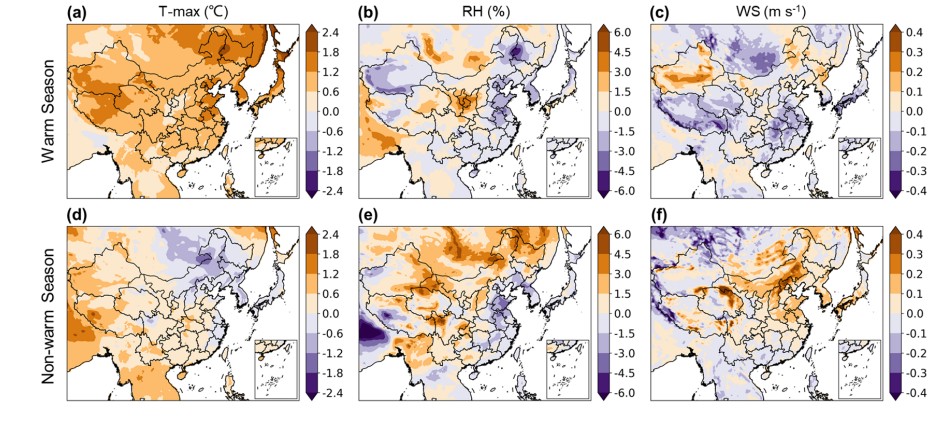





**Figure 3**

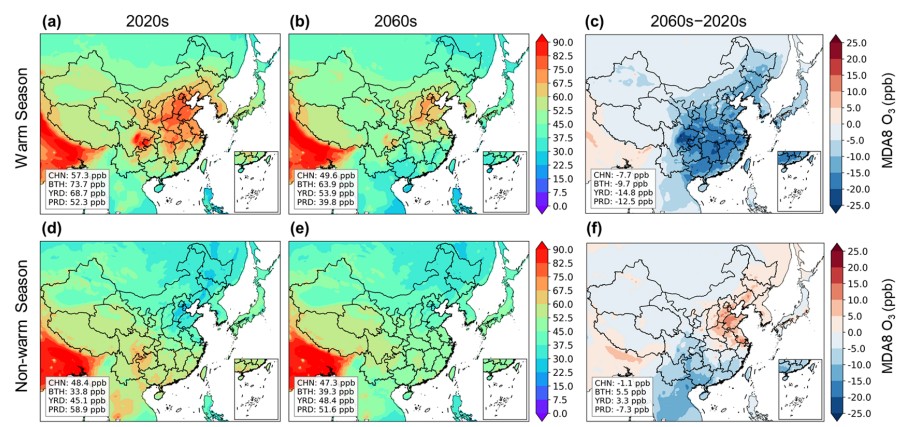



**Figure 4**

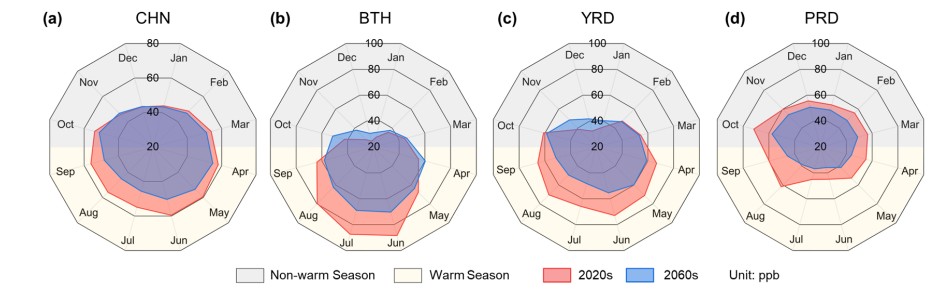





**Figure 5**

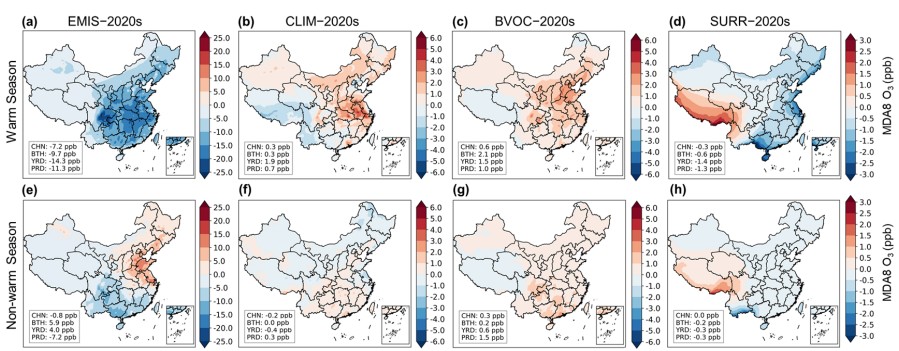




**Figure 6**

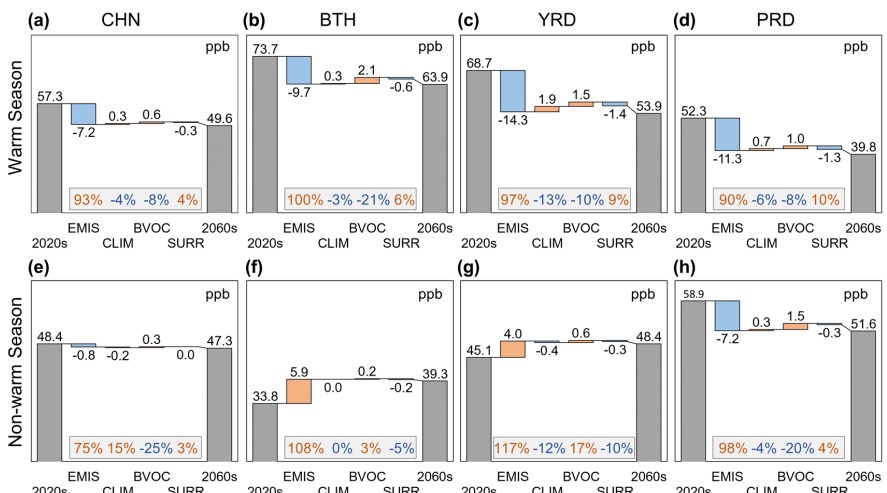





**Figure 7**

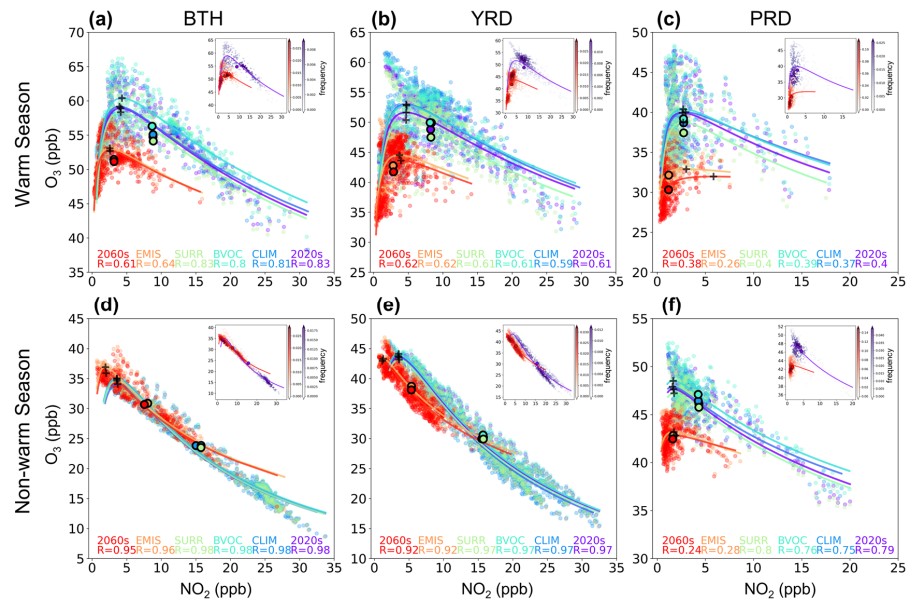




**Figure 8**

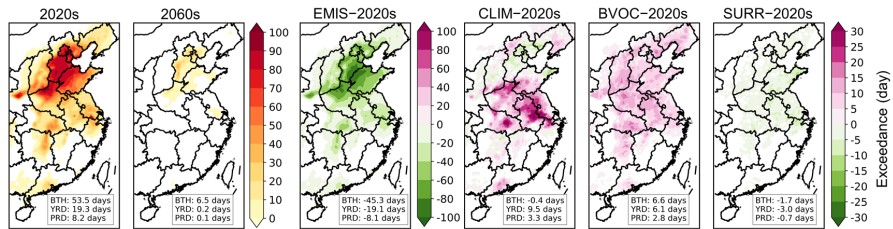
