# Peer review of "Investigating the response of China's surface ozone concentration to the future changes of multiple factors"

_EGUsphere, 2024_

## Author Response (AR1)

**Response to reviewers' comments and main revisions**

**Journal:** Atmospheric Chemistry and Physics

**Manuscript No.:** egusphere-2024-2713

**Title:** Investigating the response of China's surface ozone concentration to the future changes of multiple factors

**Authors:** Jinya Yang, Yutong Wang, Lei Zhang, Yu Zhao

We thank very much for the valuable comments and suggestions from the editor and reviewers, which help us improve our manuscript. The comments have been carefully considered and revisions have been made in response to suggestions. Following are our point-by-point responses to the comments and corresponding revisions. **Please note that the line/table/figure numbers mentioned following refer to the clean version of the revised manuscript, unless specifically noted.**

**Reviewer #1**

The authors presented a comprehensive and detailed analysis of the response of surface $O_3$ concentrations in China to future changes including factors of anthropogenic and biogenic emissions, meteorology, and transboundary transport. The topic is interesting, and the analysis is important for a better understanding of surface $O_3$ evolution in the future. I recommend this manuscript for publication, after consideration of the points below:

**Response and revisions:**

We appreciate the reviewer's positive and encouraging comment, and have made point-by-point response and revisions as summarized below.

*1.* Lines 40-43: here the authors emphasized the threat of $O_3$ pollution to human health, crop yield, and short-lived climate change. However, the references mainly focused on

crop yield and there is only one reference (Jerrett et al. 2009) for human health. It is suggested to cite a few more recent studies about human health.

**Response and revisions:**

We appreciate the reviewer's valuable comment. We have added some important references related to the human health threat from $O_3$, i.e., the global burden of disease assessment including long-term health impacted of $O_3$ (Lelieveld et al., 2015), a nationwide cohort study regarding $O_3$ and cardiovascular mortality in China (Niu et al., 2022), and the provincial estimation of $O_3$ impact on deaths, disease burden, and life expectancy (Yin et al., 2020). The information can be found in **lines 43–44 in the revised manuscript**.

*2.* Lines 114-118: recent studies (Hou et al. 2023; Liu et al. 2023) analyzed the effects of emission reduction on surface $O_3$ evolution in China in warm and non-warm seasons, which may be worth citing. Furthermore, as discussed in Lines 123-128: Hou et al. (2023) analyzed the effect of global emission reduction on surface $O_3$ evolution in China. Its conclusion seems consistent with this work.

Hou, X., Wild, O., Zhu, B., and Lee, J.: Future tropospheric ozone budget and distribution over east Asia under a net-zero scenario, Atmos. Chem. Phys., 23, 15395-15411, 10.5194/acp-23-15395-2023, 2023.

Liu, Z., Wild, O., Doherty, R. M., O'Connor, F. M., and Turnock, S. T.: Benefits of net-zero policies for future ozone pollution in China, Atmos. Chem. Phys., 23, 13755-13768, 10.5194/acp-23-13755-2023, 2023.

**Response and revisions:**

We appreciate the reviewer's valuable comment on the references. We have added the recommended references in **Lines 119–121 in the Introduction section:**

Recent studies have suggested diverse effects of future emission change on $O_3$ evolution for difference seasons in China (Hou et al. 2023; Liu et al. 2023b).

*3.* Lines 202-205: Here the description of the emission changes outside of the Chinese mainland seems inconsistent: "Emissions outside Chinese mainland are obtained from CMIP6 dataset under SSP2-45 scenario"; "emissions outside Chinese mainland are assumed the same as those in MIX Asian emission inventory".

**Response and revisions:**

We thank the reviewer for pointing out the ambiguous description. The emissions outside Chinese mainland, obtained from CMIP6 dataset, are provided as annual total at region or country level. To make them applicable for CMAQ simulation, we further downscaled the emission estimates into the gridded monthly data based on the spatial and temporal distributions of emissions in MIX Asian emission inventory. We have revised the description in **Lines 205–209 in the revised manuscript.**

*4.* Figure 3: The increase in $O_3$ concentration over the BTH in the non-warm season is interesting. I found it is consistent with the modeled results of Hou et al. 2023 and Liu et al. 2023. It is suggested to cite these studies to support the simulation in this work.

**Response and revisions:**

We appreciate the reviewer's valuable comment. We have cited the references and added relevant information in **Lines 417–420 in the Results and Discussions section:**

Similarly, Hou et al. (2023) and Liu et al. (2023b) also predicted a growth of $O_3$ concentration in non-warm season over BTH and YRD under a net-zero carbon emission scenario, resulting from a weakened titration effect.

*5.* Figure 7: The modeled $O_3$ chemical regime evolution in the 2020s is very interesting. It seems consistent with recent observation-based analysis (Chen et al. 2023; Kong et al. 2024). However, it is surprising that the $O_3$ chemical regime is still around the turning point in BTH in the 2060s. Considering the rapid cross of the turning point in the US in 1990-2010, what policy suggestions can the authors give?

Chen, X., Wang, M., He, T. L., Jiang, Z., Zhang, Y., Zhou, L., Liu, J., Liao, H., Worden, H., Jones, D., Chen, D., Tan, Q., and Shen, Y.: Data- and Model-Based Urban O3 Responses to NOx Changes in China and the United States, J. Geophys. Res.-Atmos., 128, e2022JD038228, 10.1029/2022jd038228, 2023.

Kong, L., Song, M., Li, X., Liu, Y., Lu, S., Zeng, L., and Zhang, Y.: Analysis of China's PM(2.5) and ozone coordinated control strategy based on the observation data from 2015 to 2020, J Environ Sci (China), 138, 385-394, 10.1016/j.jes.2023.03.030, 2024.

**Response and revisions:**

We thank the reviewer's comment the helpful references.

First, we have cited these observation-based references for supporting our model-based findings on the $O_3$ chemical regime in the 2020s in **lines 520–521 in the revised manuscript**.

Second, according to Chen et al. (2021), the $O_3$ chemical regime in BTH in 2019 was similar with that of northeast US (USNE) in the 1990s, while the regime of USNE had shifted to $NO_x$-limited in the 2010s. To explore the potential for a similar shift in BTH, we compared changes in anthropogenic $O_3$ precursor emissions between 1990–2010 for USNE and 2020–2060 for BTH. As shown in Table R1, the emissions of $NO_x$ and NMVOCs in 1990 USNE were comparable to those in 2020 BTH. During 1990-2010, emissions of $NO_x$ and NMVOCs were reduced by 57 % and 67 % for USNE, while the declines are predicted to be 57 % and 36 % during 2020–2060 for BTH under SSP2-45 scenario. These findings suggest that the more substantial reduction in NMVOCs in the USNE facilitated the rapid cross of the turning point in the USNE. In comparison, the weaker reduction in NMVOCs projected for BTH may hinder a similar shift. To accelerate the shift in the $O_3$ chemical regime for BTH, more ambitious reductions in NMVOCs will be necessary (ideally double the current projected abatement under SSP2-45). We have added the discussion in **lines 543–551 in the revised manuscript**.

**Table R1 The emissions and relative changes of O₃ precursors between 1990 and 2010 for USNE and those between 2020 and 2060 for BTH.**

| Emission* (Mt) or rate (%) | NO$_x$_USNE | NO$_x$_BTH | NMVOCs_USNE | NMVOCs_BTH |
|---|---|---|---|---|
| 1990USNE**/2020BTH | 4.2 | 2.2 | 4.4 | 2.2 |
| 2010USNE/2060BTH | 3.0 | 0.9 | 2.8 | 1.4 |
| Change rate | -57 % | -57 % | -67 % | -36 % |

* Emissions illustrated are obtained from USEPA for USNE and MEIC and DPEC for BTH.

** The northeast US consists of 11 states: Connecticut, Delaware, Maine, Massachusetts, Maryland, New Hampshire, New Jersey, New York, Pennsylvania, Rhode Island, and Vermont.

*6.* Lines 421-422: It may not be good to have a paragraph with only one sentence.

**Response and revisions:**

We thank and agree the reviewer's comment. We have merged the sentence into the following paragraph, and the revision is shown in **line 438 in the revised manuscript**.

*7.* Table S1: the values of simulated O₃ concentrations seem incorrect.

**Response and revisions:**

We thank the reviewer's reminder and we are sorry for the mistake. We have reviewed the entire manuscript and revised the simulated O₃ concentrations in **Table S1 in the revised supplement**.

**References**

Chen, X., Jiang, Z., Shen, Y., Li, R., Fu, Y., Liu, J., Han, H., Liao, H., Cheng, X., Jones, D. B. A., Worden, H., and Abad, G. G.: Chinese Regulations Are Working—Why Is Surface Ozone Over Industrialized Areas Still High? Applying Lessons from Northeast US Air Quality Evolution, Geophysical Research Letters, 48, e2021GL092816, 10.1029/2021gl092816, 2021.

Lelieveld, J., Evans, J. S., Fnais, M., Giannadaki, D., and Pozzer, A.: The contribution of outdoor air pollution sources to premature mortality on a global scale, Nature, 525,

367-371, 10.1038/nature15371, 2015.

Niu, Y., Zhou, Y., Chen, R., Yin, P., Meng, X., Wang, W., Liu, C., Ji, J. S., Qiu, Y., Kan, H., and Zhou, M.: Long-term exposure to ozone and cardiovascular mortality in China: a nationwide cohort study, The Lancet Planetary Health, 6, e496-e503, 10.1016/s2542-5196(22)00093-6, 2022.

Yin, P., Brauer, M., Cohen, A. J., Wang, H., Li, J., Burnett, R. T., Stanaway, J. D., Causey, K., Larson, S., Godwin, W., Frostad, J., Marks, A., Wang, L., Zhou, M., and Murray, C. J. L.: The effect of air pollution on deaths, disease burden, and life expectancy across China and its provinces, 1990–2017: an analysis for the Global Burden of Disease Study 2017, The Lancet Planetary Health, 4, e386-e398, 10.1016/s2542-5196(20)30161-3, 2020.

**Reviewer #3**

This study explores the response of surface ozone concentration in China to the future changes of multiple factors (domestic and foreign anthropogenic emissions, biogenic emissions, and meteorological conditions) under SSP2-45 scenario, based on a series of sensitivity experiments with WRF-MEGAN-CMAQ simulations. The combined and individual effects are quantified and discussed with informative figures. In general, this study is well-structured, with the results well supported by the analyses, and is well written. The manuscript can be improved by addressing the following comments.

**Response and revisions:**

We thank the reviewer's positive and encouraging comment, and have made point-by-point response and revisions as summarized below.

*1.* Section 2.3: Using the five-year mean to represent present and future scenarios is commendable. Have the authors noted any significant interannual variability within these five-year simulations that should be discussed?

**Response and revisions:**

We appreciate the reviewer's valuable comment and agree that the discussion of interannual variability within these five-year simulations was insufficiently quantified in the original submission. To address this, we use the coefficient of variation (CV) to evaluate interannual variability within the five-year simulations. CV indicates the consistency of a parameter over time and has been adopted in many multi-year simulations (Nagashima et al., 2010; Chen et al., 2019). The value is calculated with Eq. (R1), and a higher CV indicates greater interannual variability in simulated $O_3$ concentrations:

$$CV = \frac{\sigma}{\mu} \tag{R1}$$

where $\sigma$ and $\mu$ are the standard deviation and mean of the simulated $O_3$ concentrations in each five-year simulation, respectively.

As shown **in a newly added Table S3 in the revised supplement**, the CVs are generally below 5 % in most cases, indicating a low interannual variability in $O_3$ concentration simulation. This justifies the representativeness of the five-year mean for present and future scenarios. We have expanded the evaluation of interannual variability in **lines 274–279 in the revised manuscript**.

**Table S3 in the revised supplement. Coefficients of variation (CVs) for simulated $O_3$ concentrations within five-year simulations for the whole country (CHN) and selected developed regions (BTH, YRD, and PRD).**

| Simulation | Warm Season | | | | Non-warm Season | | | |
|---|---|---|---|---|---|---|---|---|
| | CHN | BTH | YRD | PRD | CHN | BTH | YRD | PRD |
| 2020s | 2.6 % | 5.7 % | 1.8 % | 4.0 % | 1.4 % | 0.9 % | 2.7 % | 3.8 % |
| 2060s | 1.5 % | 4.0 % | 2.2 % | 5.6 % | 0.8 % | 1.3 % | 2.1 % | 3.4 % |
| CLIM | 1.9 % | 5.9 % | 2.5 % | 6.5 % | 1.1 % | 1.6 % | 4.1 % | 3.8 % |
| EMIS | 2.2 % | 3.3 % | 2.2 % | 3.7 % | 1.4 % | 1.4 % | 1.5 % | 2.5 % |
| BVOC | 2.6 % | 5.0 % | 2.0 % | 4.3 % | 1.4 % | 1.0 % | 2.3 % | 3.4 % |
| SURR | 2.7 % | 5.8 % | 1.7 % | 4.1 % | 1.5 % | 0.9 % | 2.7 % | 3.9 % |

*2.* Line 229: Could you please specify which surrounding areas are considered in the analysis?

**Response and revisions:**

We thank the reviewer's reminder. Surrounding areas refer to areas other than mainland China in the modelling domain, and we have added the description in **line 233 in the revised manuscript**.

*3.* Figure 2: What causes the difference in wind speed between the warm season and non-warm season?

**Response and revisions:**

We thank the reviewer's comment. Generally, the decreasing wind speed in future

East Asia could be attributed to weakened atmospheric circulation. The "polar amplification effect" of global warming will cause a weakened equator-to-pole thermal gradient and a consequent weaker mid-latitude circulation (Coumou et al., 2018; Deng et al., 2021). The increasing wind speed in non-warm season might result from the temperature and pressure gradients between the land and adjacent oceans. The decreasing temperature in northeast China (Figure 2d), associated with the warming ocean, will enlarge the land-ocean pressure gradients and thus result in a higher wind speed in the non-warm season (Yao et al., 2019; Wu et al., 2020). Furthermore, the seasonal difference is also related to the potential future changes in atmospheric dynamics and precipitation, and the model uncertainty in model physical parameterization schemes (Kusunoki and Arakawa, 2015; Zha et al., 2020). We have added the discussion on the seasonal difference of wind speed in **lines 297–301 in the revised manuscript**.

**4.** Table S4: Can you explain what the colors represent?

**Response and revisions:**

We thank the reviewer's comment. In Figure S4, each two rows, covered by similar colors, represent the 2020s and 2060s simulation for a specific area. For each area, we highlighted the most polluted six months in one year with a darker color, to represent the change of $O_3$ pollution season between the 2020s and 2060s. To make it clearer, we change the font color of each area to match the corresponding rows in **Figure S4 in the revised Supplement**.

**5.** Conclusions and discussions: Some studies have indicated that soil NOx emissions from agricultural activities will become increasingly important in ozone formation, especially as NOx emissions from fuel combustion decrease. It is unclear how soil NOx emissions are addressed in this study. How might these emissions influence the ozone projections conducted here?

**Response and revisions:**

We thank the reviewer's comment and agree that soil $NO_x$ emissions could be important for $O_3$ formation. In this study, we did not include future change of soil $NO_x$ emissions as one factor in the research framework. Previous studies have highlighted substantial soil $NO_x$ emissions in northern China (Lu et al., 2021; Huang et al., 2023), which is also the region with great reduction in anthropogenic $NO_x$ but relatively small decline in $O_3$ concentrations in our prediction. Furthermore, soil $NO_x$ emissions are expected to increase under a warming climate, and will thus present an additional challenge for $O_3$ pollution alleviation in northern China (Wu et al., 2008; Xie et al., 2017). We have discussed the limitation in **lines 636–642 in the revised manuscript**.

**References**

Chen, Z., Zhuang, Y., Xie, X., Chen, D., Cheng, N., Yang, L., and Li, R.: Understanding long-term variations of meteorological influences on ground ozone concentrations in Beijing During 2006-2016, Environmental Pollution, 245, 29-37, 10.1016/j.envpol.2018.10.117, 2019.

Coumou, D., Di Capua, G., Vavrus, S., Wang, L., and Wang, S.: The influence of Arctic amplification on mid-latitude summer circulation, Nature Communications, 9, 10.1038/s41467-018-05256-8, 2018.

Deng, H., Hua, W., and Fan, G.: Evaluation and Projection of Near-Surface Wind Speed over China Based on CMIP6 Models, Atmosphere, 12, 10.3390/atmos12081062, 2021.

Huang, L., Fang, J., Liao, J., Yarwood, G., Chen, H., Wang, Y., and Li, L.: Insights into soil NO emissions and the contribution to surface ozone formation in China, Atmospheric Chemistry and Physics, 23, 14919-14932, 10.5194/acp-23-14919-2023, 2023.

Kusunoki, S. and Arakawa, O.: Are CMIP5 Models Better than CMIP3 Models in Simulating Precipitation over East Asia?, Journal of Climate, 28, 5601-5621, 10.1175/jcli-d-14-00585.1, 2015.

Lu, X., Ye, X., Zhou, M., Zhao, Y., Weng, H., Kong, H., Li, K., Gao, M., Zheng, B., Lin, J., Zhou, F., Zhang, Q., Wu, D., Zhang, L., and Zhang, Y.: The underappreciated role of agricultural soil nitrogen oxide emissions in ozone pollution regulation in North

China, Nat Commun, 12, 5021, 10.1038/s41467-021-25147-9, 2021.

Nagashima, T., Ohara, T., Sudo, K., and Akimoto, H.: The relative importance of various source regions on East Asian surface ozone, Atmospheric Chemistry and Physics, 10, 11305-11322, 10.5194/acp-10-11305-2010, 2010.

Wu, J., Shi, Y., and Xu, Y.: Evaluation and Projection of Surface Wind Speed Over China Based on CMIP6 GCMs, Journal of Geophysical Research: Atmospheres, 125, 10.1029/2020jd033611, 2020.

Wu, S., Mickley, L. J., Jacob, D. J., Rind, D., and Streets, D. G.: Effects of 2000-2050 changes in climate and emissions on global tropospheric ozone and the policy-relevant background surface ozone in the United States, Journal of Geophysical Research-Atmospheres, 113, 10.1029/2007jd009639, 2008.

Xie, M., Shu, L., Wang, T.-j., Liu, Q., Gao, D., Li, S., Zhuang, B.-l., Han, Y., Li, M.-m., and Chen, P.-l.: Natural emissions under future climate condition and their effects on surface ozone in the Yangtze River Delta region, China, Atmospheric Environment, 150, 162-180, 10.1016/j.atmosenv.2016.11.053, 2017.

Yao, Y., Zou, X., Zhao, Y., and Wang, T.: Rapid Changes in Land-Sea Thermal Contrast Across China's Coastal Zone in a Warming Climate, Journal of Geophysical Research: Atmospheres, 124, 2049-2067, 10.1029/2018jd029347, 2019.

Zha, J., Wu, J., Zhao, D., and Fan, W.: Future projections of the near-surface wind speed over eastern China based on CMIP5 datasets, Climate Dynamics, 54, 2361-2385, 10.1007/s00382-020-05118-4, 2020.